# Maternal Metal Ion Status Along Pregnancy and Perinatal Outcomes in a Group of Mexican Women

**DOI:** 10.3390/ijms252313206

**Published:** 2024-12-08

**Authors:** Johana Vásquez-Procopio, Johnatan Torres-Torres, Elodia Rojas-Lima, Salvador Espino-y-Sosa, Juan Mario Solis-Paredes, Maribel Sánchez-Martínez, Mari-Cruz Tolentino-Dolores, Otilia Perichart-Perera, Fanis Missirlis, Guadalupe Estrada-Gutierrez

**Affiliations:** 1Department of Basic Sciences, Instituto Tecnológico del Valle de Oaxaca, Tecnológico Nacional de México, Santa Cruz Xoxocotlán 71233, Mexico; johana.vp@voaxaca.tecnm.mx; 2Deparment of Immunobiochemistry, Instituto Nacional de Perinatología Isidro Espinosa de los Reyes, Mexico City 11000, Mexico; maribel71sm@yahoo.com.mx; 3Department of Reproductive and Perinatal Health Research, Instituto Nacional de Perinatología Isidro Espinosa de los Reyes, Mexico City 11000, Mexico; torresmmf@gmail.com (J.T.-T.); juan.mario.sp@gmail.com (J.M.S.-P.); 4Unidad de Investigación de Salud en el Trabajo, Centro Médico Nacional Siglo XXI, Instituto Mexicano del Seguro Social, Mexico City 06720, Mexico; elodia.rojas.lima@gmail.com; 5Research Division, Instituto Nacional de Perinatología Isidro Espinosa de los Reyes, Mexico City 11000, Mexico; salvadorespino@gmail.com; 6Department of Nutrition and Bioprogramming, Instituto Nacional de Perinatología Isidro Espinosa de los Reyes, Mexico City 11000, Mexico; cruz_tolentino@yahoo.com.mx (M.-C.T.-D.); oti_perichart@yahoo.com (O.P.-P.); 7Department of Physiology, Biophysics and Neuroscience, Center for Research and Advanced Studies (Cinvestav), Mexico City 07360, Mexico; fanis@fisio.cinvestav.mx

**Keywords:** pregnancy, maternal mineral status, perinatal outcomes

## Abstract

Pregnancy increases the demand for essential metal ions to support fetal development, making the maternal metal ion status a critical determinant of perinatal outcomes. This prospective cohort study examined changes in metal ion levels across the three trimesters, evaluated the influence of preexisting metabolic conditions on the metal ion status, and assessed the associations between maternal metal ion levels and perinatal outcomes in 206 pregnant women from the Biochemical and Epigenetic Origin of Overweight and Obesity (OBESO) cohort receiving care at the Instituto Nacional de Perinatología in Mexico City from 2017 to 2020. Six essential metal ions (iron, zinc, copper, calcium, magnesium, and phosphorus) were measured in blood samples using inductively coupled plasma optic emission spectrometry. Significant variations in the metal ion levels were observed across the trimesters, with notable decreases in iron and magnesium and increases in copper as pregnancies progressed. Maternal hypothyroidism was associated with significantly low levels of zinc and magnesium during pregnancy. Regression analyses revealed robust associations between maternal metal ion levels and perinatal outcomes. For instance, declining magnesium levels as pregnancies progressed were positively associated with gestational diabetes (OR: 2.92, *p* = 0.04; OR: 2.72, *p* = 0.03). The maternal metal ion status significantly influences perinatal outcomes.

## 1. Introduction

Malnutrition and nutrient deficiencies during pregnancy remain significant public health concerns, even in developed nations [1,2]. Optimal nutrient levels are crucial for maintaining maternal health and supporting fetal growth and development [3]. Iron (Fe) deficiency stands out as the most common nutritional deficiency during pregnancy [4]. In mothers, it is associated with reduced physical and cognitive performance, increased fatigue, higher susceptibility to infections, greater risk of hospitalization, and inhibited lactation [5]. For the fetus, adverse outcomes include spontaneous abortion, preterm delivery, intrauterine fetal death, low birth weight, small for gestational age infants, hypertension, and neurological impairments [6,7]. After Fe, zinc (Zn) is the second most vital element in the human body [8]. Maternal Zn deficiency during pregnancy may have a direct effect on fetal growth and increase the risk of a low birth weight, preterm delivery, postpartum hemorrhage, and congenital malformations [3]. Similarly, copper (Cu) plays a vital role in maternal and fetal health. Although previously considered rare, Cu deficiency has been increasingly reported, and it is associated with complications such as ruptured membranes, preterm birth, and stress-related fetal growth restriction [9]. Low calcium (Ca) levels during pregnancy are linked to hypertensive disorders [10,11], and they can increase the risk of fetal growth abnormalities and preeclampsia, one of the leading causes of maternal mortality in Mexico and Latin America [12,13]. Magnesium (Mg) deficiency is associated with restricted fetal growth, intrauterine growth restriction, preterm labor, preeclampsia, and especially gestational diabetes [14,15]. Finally, phosphorus (P) is essential during pregnancy; however, deficiencies are rare in pregnant women [1].

Additional factors contributing to adverse perinatal outcomes include maternal age and chronic conditions such as hypertension, diabetes mellitus, being overweight, and obesity [7,15,16,17,18]. These conditions are often associated with altered lipid profiles and glucose metabolism, which can disrupt metal ion homeostasis [19].

Despite their importance, prospective investigations into metal ion concentrations throughout the stages of pregnancy are scarce [20,21,22,23], and there are limited studies which explore the relationship between essential element levels during pregnancy and perinatal outcomes [24,25,26].

A longitudinal assessment of essential element statuses during pregnancy and their influence on perinatal outcomes is therefore critical for mitigating risks, facilitating early diagnosis, and ensuring timely maternal healthcare [27,28,29,30,31,32]. Insights from such studies can inform the development of region-specific dietary and supplementation strategies tailored to the unique needs of pregnant individuals, helping to reduce treatment costs and improve perinatal outcomes. The primary objective of this study was to evaluate the whole blood concentrations of six essential metal ions (Fe, Zn, Cu, Ca, Mg, and P) and describe their trends throughout pregnancy. Additionally, we aimed to assess the association between blood changes in these metal ion levels and preexisting maternal metabolic conditions and identify key windows of susceptibility during which metal ions significantly affect perinatal outcomes.

## 2. Results

### 2.1. Description of the Cohort and Characteristics of the Study Population

A cohort of 206 pregnant women from the OBESO project in Mexico City was studied from 2017 to 2020. The baseline and biochemical characteristics of the cohort are summarized in Table 1, with detailed information provided in Appendix A. Various comorbidities, such as systemic arterial hypertension (2%), diabetes mellitus (5%), and hypothyroidism (22%), were observed among the study population. The mean maternal age was 30 years (±5.3), and 61% were overweight or obese. The participants predominantly belonged to the low socioeconomic status group (64%) followed by the middle class (34%), while those in the high-class group were minimal (2%). Multivitamin supplementation was reported in 64% of participants during the first trimester and increased to 85% in the second and third trimesters. Vitamin D supplementation data revealed that 48% of the participants were supplemented during the first trimester, followed by 70% during the second trimester and 73% during the third trimester. The metabolic parameters significantly varied across the trimesters, including the triglycerides, total cholesterol, HDL cholesterol, LDL cholesterol, and 25-OH vitamin D.

### 2.2. Changes in Metal Ion Levels Across Trimesters

Maternal metal ion levels (Fe, Zn, Cu, Ca, Mg, and P) were compared across trimesters (Figure 1), showing significant decreases in Fe and Mg in the third and second trimesters compared with the first (*p* < 0.0001). Zn levels notably decreased in the second (*p* < 0.001) and third trimesters (*p* < 0.0001), while low P levels were observed in the second (*p* < 0.0001) and third trimesters (*p* = 0.0004). Cu levels increased as the pregnancy progressed (*p* < 0.0001), and Ca levels increased only in the second trimester versus the first (*p* = 0.0432). The Cu/Zn ratio was calculated and showed a significant increase in the third trimester compared with the first (*p* < 0.0001) and second trimesters (*p* < 0.0001) (Appendix A).

### 2.3. Influence of Preexisting Metabolic Conditions on Metal Ion Status

Mixed-effects analyses explored the interactions between preexisting metabolic conditions and metal ion statuses (Appendix A). Zn levels significantly varied across the trimesters, being influenced by hypothyroidism diagnoses (*p* = 0.035) (Figure 2). Similarly, Mg levels showed significant variation associated with hypothyroidism (*p* = 0.014). However, other conditions and pregestational body mass index (pBMI) categories, while showing patterns of change, did not reach statistical significance (Appendix A).

### 2.4. Associations Between Metal Ions and Perinatal Outcomes

Major perinatal outcomes included a low birth weight (12%), neonatal admission to the intensive care unit (ICU) (11%), preterm birth (11%), preeclampsia (9%), gestational diabetes (7%), obstetric hemorrhage (5%), and respiratory distress syndrome (1%) (Table 2). Individual and combined associations of metal ions with perinatal outcomes were examined to identify windows of susceptibility (Table 3 and Figure 3).

Low Mg levels in the first trimester were associated positively with gestational diabetes (OR: 1.29, *p* = 0.03) and P levels in the second (OR: 1.59, *p* = 0.04) and third trimesters (OR: 1.76, *p* = 0.02) (Table 3). Zn, Mg, and P were positive and significantly associated with ICU admission in the first trimester (OR: 1.06, *p* < 0.01; OR: 1.21, *p* = 0.04; OR: 1.68, *p* = 0.01, respectively), while calcium (Ca) levels were linked to ICU admission in the second and third trimesters (OR: 1.29, *p* = 0.02; OR: 1.28, *p* = 0.02, respectively). Conversely, the Cu/Zn ratio was negatively associated with ICU admission in the first trimester. Zn levels in the first trimester were positively associated with a low birth weight (OR: 1.03, *p* = 0.03) and preterm birth (OR: 1.04, *p* = 0.009), alongside Mg (OR: 1.19, *p* = 0.04). However, the Cu/Zn ratio showed a significant protective effect for this outcome (OR: 0.63, *p* = 0.03). Additionally, Ca levels in the third trimester were associated positively and significantly with a low birth weight (OR: 1.19, *p* = 0.03).

In the first trimester, the mixed-effects analysis showed that the P levels were negative and significantly associated with low birth weights (PIP = 0.7354) (Figure 3M, Appendix A), while the Zn levels had the opposite effect. In the third trimester, Ca levels were positively associated with ICU admission (PIP = 0.9194) (Figure 3L, Appendix A).

Additional associations between perinatal outcomes and metal ions reached significance in at least one quartile (q0.25, q0.5, or q0.75) across all three trimesters, indicating either a negative or positive association. For instance, the Cu levels in at least one quartile were negative and significantly associated with preeclampsia in the first and third trimesters (Figure 3B). Similarly, Cu showed a protective effect against neonatal ICU admission in the third trimester (PIP = 0.8158) (Figure 3B, Appendix A).

Furthermore, low Fe levels throughout pregnancy were positively associated with obstetric hemorrhage (Figure 3G–M) and preterm birth (Figure 3Q), while low P levels demonstrated protective effects.

### 2.5. Longitudinal Analysis

Longitudinal analyses identified associations between specific metal ion levels and adverse perinatal outcomes (Table 4). For instance, low Zn, P, and Mg levels throughout pregnancy were positively associated with developing gestational diabetes (OR: 2.43, *p* = 0.04; OR: 1.97, *p* = 0.04; OR: 2.72, *p* = 0.03, respectively). Additionally, low Mg levels in the first trimester showed a similar positive association (OR: 2.61, *p* = 0.029). In the first trimester, Zn, P, and Ca levels were also significantly associated with an increased risk of neonatal ICU admission (OR: 1.93, *p* = 0.04; OR: 1.73, *p* = 0.04; OR: 1.84, *p* = 0.02, respectively).

## 3. Discussion

In this prospective cohort study, the observed variations in the levels of Cu, Fe, Zn, Mg, and P throughout pregnancy support previous findings from studies conducted in Korean and Chinese [28,32]. These variations highlight the significant impact of pregnancy on metal ion homeostasis, driven by the increased demands of fetal development. Specifically, Cu absorption increases during pregnancy due to the greater need for Cu-dependent enzymes, such as cytochrome c oxidase, which is essential for aerobic respiration, and superoxide dismutases, the enzymes which catalyze the dismutation of superoxides into oxygen and hydrogen peroxide [33]. Increased Cu levels during pregnancy interfere with the absorption of Zn and thus explain the low concentration of Zn [3]. Zn is widely recognized for its critical roles in cell division, differentiation, and function, which are essential for tissue growth [8]. Low Fe levels are common in pregnant woman because the volume of blood increases to supply oxygen to the baby. Fe is necessary for the synthesis of oxygen and transport proteins, particularly hemoglobin and myoglobin, or for the formation of heme enzymes and other iron-containing enzymes involved in electron transfer and oxidation reductions [33]. Mg and P are cations pivotal to many biochemical and physiological processes, being involved in many biological and cellular functions, including protein synthesis and nucleotide metabolism [14]. Some studies have shown that Mg and P levels decline as pregnancies advance but have also reported improvements in their statuses, with supplementation leading to significant improvements in maternal and perinatal outcomes [34].

Several preexistent maternal factors contributing to adverse perinatal outcomes disrupting mineral homeostasis, such as Ca, Mg, and P levels, are frequently associated with thyroid dysfunction [35]. Hypothyroidism is prevalent in pregnant women, especially in a developing country, increasing the potential for maternal and fetal adverse outcomes [36]. Our study elucidates this phenomenon by revealing significantly low Zn and Mg levels in pregnant women diagnosed with hypothyroidism, highlighting the intricate interplay between cation metabolism and thyroid hormones. These results are consistent with previous studies which reported decreases in Zn and Mg concentrations in patients with hypothyroidism [37,38,39].

The associations between maternal metal ion levels and perinatal outcomes observed in this study are particularly important in regions like Latin America [40]. We observed a significant positive association between low Mg and P levels and gestational diabetes. This association was consistent across longitudinal models and trimester-specific windows of susceptibility analyses. Mg is essential for carbohydrate metabolism and may influence hormonal activity involved in glucose regulation, highlighting its role in the pathogenesis of gestational diabetes [41,42]. Although studies have consistently reported associations between hypomagnesemia and metabolic disorders, including gestational diabetes, the exact role of Mg dysregulation in disease onset remains unclear, necessitating further investigation [43,44,45,46,47,48,49,50]. Phosphorus also plays a critical role in the intermediate metabolism of carbohydrates [51]. Diabetes mellitus, a chronic metabolic disorder characterized by impaired glucose metabolism, is associated with alterations in serum metal ion levels, including P, Mg, and Zn [47,52]. However, no significant associations between P and pregnant women with gestational diabetes have been reported [53]. Nevertheless, in cases of severe, uncontrolled diabetes mellitus, elevated blood sugar levels can lead to low phosphate levels due to intracellular glucose phosphorylation [54].

Similarly, our study highlights a positive association between low Zn levels and gestational diabetes, with notable effects observed in the second and third trimesters. Zn is essential for the synthesis, packaging, and secretion of insulin in pancreatic β cells [55]. The Zn transporter ZnT8 plays a critical role in this process; its experimental deletion or the presence of certain genetic variants has been linked to a higher risk of developing diabetes [56]. Gestational diabetes is a complex metabolic disorder with significant implications for maternal and fetal health. Despite advances in understanding its underlying mechanisms, effective prevention and treatment strategies remain limited [57]. Therefore, supplementation with metal ions such as Zn during pregnancy could potentially reduce the incidence of gestational diabetes, which is particularly prevalent among Latin American women [40]. Moreover, the essential role of Zn as a cofactor in numerous biological processes emphasizes the importance of monitoring its levels during pregnancy to reduce complications.

Our study reaffirms that optimal levels of essential metal ions, such as Ca, Mg, and Zn, are critical in preventing adverse perinatal outcomes, including neonatal ICU admission, low birth weights, and preterm birth. An increase in the Cu/Zn ratio during pregnancy was negatively associated with ICU admission and preterm delivery. These findings corroborate that, in some cases, the Cu/Zn ratio may serve as a better indicator than the independent measurement of these ions [58]. Additionally, low Mg concentrations early in pregnancy were positively associated with preeclampsia, whereas Cu exhibited a protective effect. These results emphasize the importance of adequate supplementation to mitigate preeclampsia, the second leading cause of maternal mortality in Mexico and Latin America. Mg plays a key role in energy-dependent metabolic processes and significantly affects cardiac excitability, vascular tone, contractility, and reactivity [40]. Hypomagnesemia in pregnant women is commonly associated with physiological changes, including hemodilution, increased renal clearance, and mineral consumption by the growing fetus [59]. Mg levels tend to decrease with advancing gestational age during normal pregnancy, and disturbances in Mg homeostasis have been observed in women who later developed preeclampsia [15,30].

These findings have significant implications for obstetric and perinatal care. Incorporating assessments and supplementation of Mg, Ca, Zn, Fe, and P levels into prenatal care protocols can aid clinicians in identifying women at higher risk of complications such as gestational diabetes, preeclampsia, newborn ICU admission, and low birth weights in different countries in Latin America and the Caribbean, where the prevalence of these outcomes is high [40]. Furthermore, the observed protective effects of specific metal ions, such Ca, Cu and Zn, against adverse outcomes suggest potential preventive interventions, as previously reported by other authors as well [12,28]. Strategies aimed at addressing deficient metal ion levels through supplementation or dietary modifications may help mitigate the risk of complications such as preeclampsia and newborn ICU admissions [60,61]. Hypothesized mechanisms underlying these associations include the role of Mg in regulating blood pressure and endothelial function. At the same time, the involvement of Mg in carbohydrate metabolism and hormonal regulation may explain its association with gestational diabetes. The associations between maternal metal ion levels and perinatal outcomes suggest promising avenues for future research to elucidate the underlying mechanisms driving these associations. Conducting clinical trials to understand the biological pathways through which metal ions influence pregnancy outcomes could pave the way for developing novel therapeutic targets and preventive strategies [27,29].

Longitudinal studies are essential for establishing causal relationships between maternal metal ion statuses and perinatal outcomes. These studies should carefully account for potential confounding factors such as maternal age, BMI, and dietary habits. Additionally, prospective cohort studies with larger sample sizes are warranted to provide more robust evidence and validate the observed associations across diverse populations. Moreover, intervention studies are needed to assess the efficacy and safety of targeted strategies to modulate maternal metal ion levels. Rigorous randomized controlled trials investigating the impact of supplementation or dietary interventions on pregnancy outcomes could offer valuable insights into potential preventive measures for complications such as gestational diabetes and preeclampsia. Such research endeavors are crucial for translating our findings into actionable strategies to improve maternal and neonatal health outcomes.

Our study exhibits several strengths which enhance the robustness and reliability of our findings. We adopted a longitudinal design, enabling us to monitor maternal metal ion levels across multiple trimesters, offering a comprehensive understanding of their dynamics throughout pregnancy. Moreover, including a diverse sample enhances the generalizability of our results, making them applicable to various demographic groups. Rigorous statistical analyses were conducted to control for potential confounding variables and minimize biases. Additionally, standardized outcome measures and clinical data collection methods were employed, bolstering the validity and reliability of our findings.

However, certain limitations should be acknowledged when interpreting our results. Despite efforts to adjust for confounding factors, residual confounding remains possible due to the observational nature of our study. Moreover, measuring maternal metal ion levels at discrete time points may not capture fluctuations between assessments. Reliance on maternal blood samples for metal ion measurements may not fully reflect their bioavailability or concentrations at the fetal–placental interface. Although our sample size was adequate for detecting significant associations, generalizability to larger populations or specific subgroups may be limited. Lastly, the observational design precludes causal inference, necessitating further interventional research to establish the causal role of maternal metal ion levels in perinatal outcomes.

## 4. Materials and Methods

### 4.1. Study Population

This prospective cohort study was conducted as part of the ongoing Biochemical and Epigenetic Origin of Overweight and Obesity (OBESO) project at the Instituto Nacional de Perinatología “Isidro Espinosa de los Reyes” in Mexico City, Mexico. OBESO is an institutional cohort of pregnant women and their children up to 2 years of age which is aimed at studying the biochemical, clinical, lifestyle, and epigenetic determinants of obesity. Women were recruited at the Department of Maternal-Fetal Medicine and assessed in a nutrition clinic during the three trimesters of pregnancy: T1 (11–13.6 weeks of gestation), T2 (18–22.6 weeks of gestation), and T3 (28–34.6 weeks of gestation). The sample was selected through convenience (January 2017–January 2020) according to the following inclusion criteria: healthy adult women (without metabolic syndromes) and those with comorbidities such as diabetes mellitus, hypertension, being overweight, and obesity. The exclusion criteria for this study included pregnant women with multiple pregnancies, renal or hepatic diseases, infectious diseases, congenital malformations, or autoimmune disorders.

### 4.2. Data Collection

Clinical data were retrieved from medical records, including maternal age, gestational age at admission, weight, and height. The pBMI was calculated and categorized according to WHO criteria: normal weight (pBMI = 18.5 to 24.9), overweight (pBMI ≥ 25), or obesity (pBMI ≥ 30). Underweight women (BMI < 18.5) were not included in this study. A multivitamin was prescribed by obstetricians or other health professionals involved in prenatal care, and the decision was independent from this study; the women were asked about supplement use. The socioeconomic status (low, medium, and high) was considered a potential confounder, and it was assessed with six dimensions of well-being within the household—human capital, practical infrastructure, connectivity and entertainment, health infrastructure, planning and future, and basic infrastructure and space—according to Mexican Association of Market Survey and Public Opinion (Asociación Mexicana de Investigación de Mercados y Opinión Pública) questionnaires and INper parameters [62].

### 4.3. Biochemical Analysis

The fasting maternal blood samples were collected in Vacutainer tubes (Becton-Dickinson, Franklin Lakes, NJ, USA) and centrifuged for 15 min at 1000× *g*. The fasting serum triglyceride, total cholesterol, low-density lipoprotein, and high-density lipoprotein concentrations were measured through enzymatic colorimetric methods using an automated analyzer (ISE Echo Lory 2000) and commercial kits (DiaSys Diagnostic Systems GmbH, Holzheim, Germany). The 25-hydroxyvitamin D concentration was found using ELISA (chemiluminescence; Architect Abbott Diagnostics, Lake Forest, IL, USA).

### 4.4. Measurement of Metals

Fasting blood samples were collected from each participant during all three trimesters of pregnancy using BD Vacutainer^®^ Blood Collection Tubes for Trace Element Testing and stored at −80 °C until analysis. Subsequently, three different volumes of blood (10, 20, and 30 µL) were transferred to Eppendorf plastic microtubes containing 200 µL of concentrated ultrapure nitric acid (Merck). After sealing, the tubes were incubated at 60 °C for 24 h for digestion, followed by dilution with 1 mL of miliQ water. Elemental analysis (Fe, Zn, Cu, Ca, Mg, and P) was conducted using a PerkinElmer Optima 8300 ICP-OES instrument (Shelton, CT, USA), employing appropriate calibration curves and a digestion blank containing ultrapure nitric acid alone [58].

### 4.5. Perinatal Outcomes

All birth outcome data were extracted from medical records. Gestational age was determined based on the last menstrual period and corrected by an ultrasound examination in the first trimester. Preterm birth was defined as delivery before 37 completed weeks of gestation, while a low birth weight was defined as a neonate birth weight of less than 2500 g, regardless of gestational age. Preeclampsia was diagnosed as the onset of hypertension (≥140/90 mmHg on two separate occasions at least 4 h apart) and proteinuria (≥300 mg/24 h) after 20 weeks of gestation [63]. Gestational diabetes was diagnosed according to the American Diabetes Association 2010 criteria based on the oral glucose tolerance test (OGTT) glucose levels measured between 24 and 28 weeks of gestation [64]. Pregnant women who presented with obstetric hemorrhage (blood loss of 500 mL or more within 24 h after birth) and newborns who were admitted to the ICU were also considered perinatal outcomes.

### 4.6. Statistical Methods

Statistical analyses were performed using Stata version 18, GraphPad Prism 10, and R version 4.3.2. Descriptive statistics included the medians and standard deviations for the continuous variables and percentages for the categorical variables.

Normality was assessed using the Kolmogorov–Smirnov test. Group comparisons were performed using McNemar test, Friedman’s test, and the Wilcoxon matched pairs signed-rank test at a significance level of *p* < 0.05. Spearman correlation analyses were employed to explore the associations between the metal ion levels and the maternal clinical and biochemical parameters on a continuous scale.

To evaluate the changes in metal ion levels influenced by preexisting conditions over each trimester of pregnancy, a linear mixed effects (LME) model with random effects and interaction terms was utilized.

Two approaches were used to assess the influence of metal ion statuses on perinatal outcomes. The first approach identified susceptibility periods by measuring the levels of each metal ion per trimester and evaluating their relationships with the outcomes. The second approach involved a longitudinal analysis to examine the association with outcomes over time. In all cases, the models were adjusted for confounders, which were selected based on the theoretical framework and included age, pBMI, glucose, triglycerides, total cholesterol, vitamin D, multivitamin supplementation, and socioeconomic status.

#### 4.6.1. Susceptibility Periods During Pregnancy

Two statistical methods were employed simultaneously to analyze the susceptibility periods. The first method used logistic regression models, where the metal ion levels measured in each trimester were included as explanatory variables and evaluated for their relationships with each outcome. Given the correlation between metal ion levels across trimesters, a mixture analysis was also conducted using Bayesian kernel machine regression (BKMR) with the “bkmr” package in R(R version 4.3.1 (16-06-2023)). This method leverages Bayesian and statistical learning techniques to regress an outcome variable on a non-parametric term of exposed mixture components using a Gaussian kernel function. For this report, the dose–response relationships between exposures and outcomes were assessed. Posterior inclusion probabilities (PIPs) were calculated for the BKMR to describe the relative importance of each exposure to the outcome. The effect of each metal ion was estimated for the 0.25, 0.50, and 0.75 quantiles.

#### 4.6.2. Longitudinal Analysis

Given the longitudinal design of this study, an LME model was used with a two-stage analysis to explore the association between the changing trends of metal ions across the three trimesters of pregnancy. In the first stage, an LME model was applied to each metal ion level as well as the chemical parameters of glucose, triglycerides, cholesterol, and vitamin D throughout pregnancies. The best linear unbiased predictors of slopes and random effects were calculated. The intercept represented the difference in the levels of each metal ion in the first trimester, and the slope represented the average change in ion levels over time during pregnancy. In the second stage, the estimates of the intercepts and slopes of the metal ions were used as predictors in logistic regression models for each outcome studied to calculate the odds ratios (ORs) and the 95% confidence intervals adjusted for confounders.

## 5. Conclusions

In conclusion, our study emphasizes the substantial influence of maternal magnesium, phosphate, calcium, zinc, and copper ion levels on perinatal outcomes, underscoring their clinical relevance and potential public health implications. These findings provide valuable insights into the complex interplay between maternal metal ion statuses and pregnancy outcomes, warranting further investigation to advance our understanding and facilitate the development of targeted interventions to improve maternal and neonatal health.

## Figures and Tables

**Figure 1 ijms-25-13206-f001:**
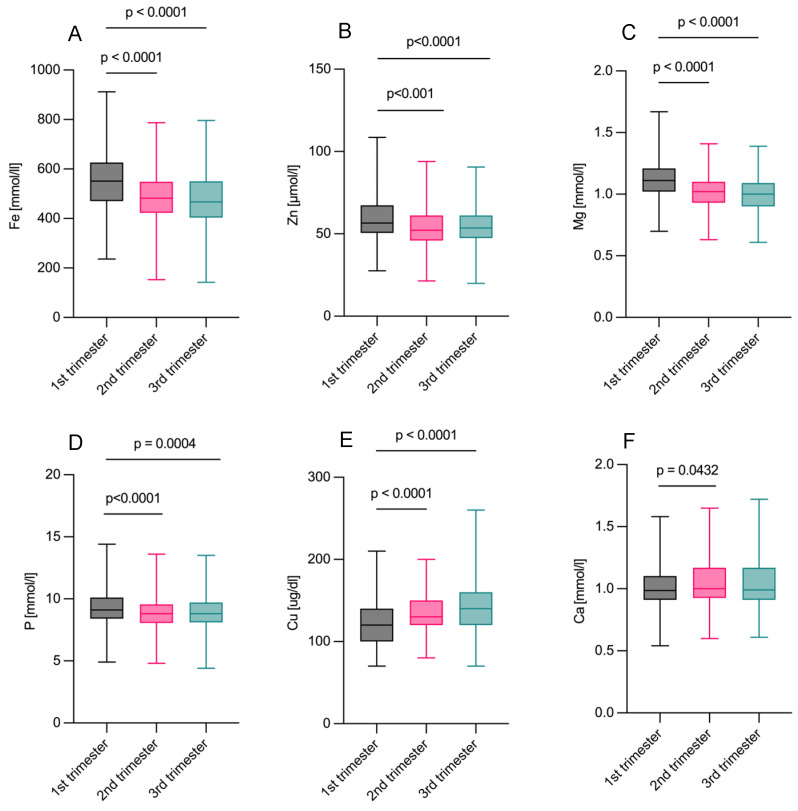
Maternal whole blood metal ion concentrations across pregnancy. (**A**–**D**) Levels of Fe, Zn, Mg, and P declined with advancing gestation. (**E**) Cu concentrations showed a significant increase during the second and third trimesters. (**F**) Ca levels increased in the second trimester versus the first trimester. Standard deviations of the means are indicated. Group comparisons were conducted using the Friedman test. Sample size: n = 206. Significance level: *p* < 0.05.

**Figure 2 ijms-25-13206-f002:**
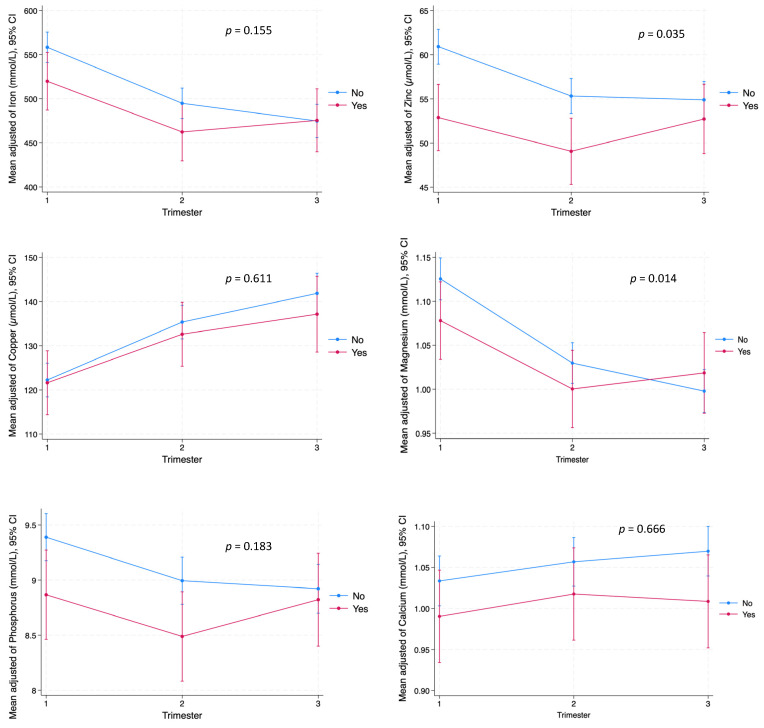
Linear mixed-effects model with interaction between hypothyroidism diagnosis and maternal metal ion levels during pregnancy. The blue line denotes healthy patients, while the red line represents those with hypothyroidism. Significant interactions were observed for Zn and Mg levels throughout pregnancy, while Fe, Cu, P, and Ca remained stable. Healthy patients: n = 161, Patients with hypothyroidism: n = 45. Significance level: *p* < 0.05.

**Figure 3 ijms-25-13206-f003:**
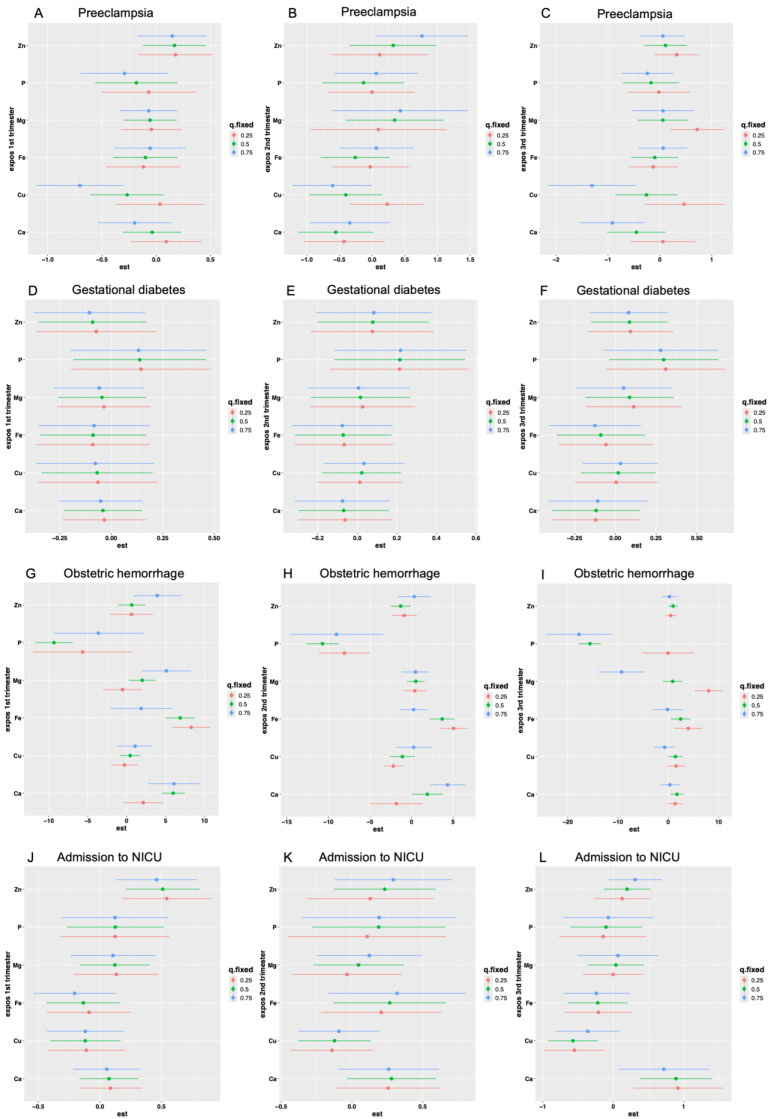
Estimated effect at 95% confidence interval in quartile of combined metal ion levels to explore windows of susceptibility to adverse perinatal outcomes. (**A**–**C**) Preeclampsia: Copper (Cu) showed a significant protective effect at the q0.75 quartile in the first trimester. In the third trimester, both Cu and calcium (Ca) exhibited protective effects at q0.75, while magnesium (Mg) showed a high-risk effect at q0.25. (**D**–**F**) Gestational diabetes: No significant effects from the metal ions were observed in any trimester. (**G**–**I**) Obstetric hemorrhage: Phosphorus (P) significantly reduced the risk across at least one quartile in all three trimesters, whereas iron (Fe) and Ca were associated with increased risk. (**J**–**L**) Neonatal intensive care admission: Zinc (Zn) was significant across all quartiles in the first trimester. In the third trimester, Ca was associated with an increased risk, while Cu reduced the risk at q0.5 and q0.75. (**M**–**O**) Low birth weight: In the first trimester, Zn was associated with an increased risk of a low birth weight, whereas P had a protective effect. (**P**–**R**) Preterm birth: In the first trimester, Cu was significant at q0.5 and q0.75, with Ca at q0.5 showing a high-risk effect. In the second trimester, Fe exhibited a high-risk effect at q0.5 and q0.75.

**Table 1 ijms-25-13206-t001:** Baseline and biochemical characteristics of the study population.

Variable	All Women n = 206	First Trimester	Second Trimester	Third Trimester	*p* Value
Maternal age, mean ± SD	30.3 ± 5.3	---	---	---	---
Pre-pregnancy BMI, mean ± SD	26.9 ± 4.8	---	---	---	---
Normal weight, n (%)	80 (39%)	---	---	---	---
Overweight, n (%)	74 (36%)	---	---	---	---
Obesity, n (%)	51 (25%)	---	---	---	---
Systemic arterial hypertension, n (%)	4 (2%)	---	---	---	---
Diabetes mellitus, n (%)	11 (5%)	---	---	---	---
Hypothyroidism, n (%)	45 (22%)	---	---	---	---
Socioeconomic status, n (%)					
High	5 (2)	---	---	---	
Medium	69 (34)	---	---	---	
Low	132 (64)	---	---	---	
Multivitamin supplementation, n (%)	---	131 (64)	175 (85)	175 (85)	<0.0001 ^a^
Vitamin D supplementation, n (%)	---	98 (48)	145 (70)	151 (73)	<0.0001 ^a^
Triglycerides (mg/dL), mean ± SD	---	133.3 ± 47.4	176.0 ± 57.8	217.1 ± 67.2	<0.0001 ^b^
Total cholesterol (mg/dL), mean ± SD	---	179.2 ± 40.8	211.9 ± 44.6	235.9 ± 50.9	<0.0001 ^b^
HDL cholesterol (mg/dL), mean ± SD	---	56.0 ± 11.5	62.3 ± 11.9	64.2 ± 14.0	<0.0001 ^b^
LDL cholesterol (mg/dL), mean ± SD	---	86.1 ± 23.2	110.3 ± 32.0	120.9 ± 34.6	<0.0001 ^b^
25-OH vitamin D, (ng/dL), mean ± SD	---	20.9 ± 6.9	24.6 ± 9.2	26.0 ± 9.4	<0.0001 ^b^

^a^ McNemar test. ^b^ Friedman test.

**Table 2 ijms-25-13206-t002:** Perinatal outcomes of the study population.

Outcomes	All Women (n = 206)
Gestational age at delivery (weeks), mean ± SD	38.1 ± 1.8
Preterm birth, n (%)	22 (11%)
Newborn birthweight (g), mean ± SD	2.97 ± 0.45
Low birthweight, n (%)	25 (12%)
Preeclampsia, n (%)	18 (9%)
Gestational diabetes, n (%)	15 (7%)
Respiratory distress syndrome, n (%)	3 (1%)
Admission to neonatal intensive care unit, n (%)	23 (11%)
Obstetric hemorrhage, n (%)	10 (5%)

**Table 3 ijms-25-13206-t003:** Individual metal ion associations with adverse perinatal outcomes.

		First Trimester		Second Trimester		Third Trimester	
Outcome	Metal	OR (95% CI)	*p* Value	OR (95% CI)	*p* Value	OR (95% CI)	*p* Value
Preeclampsia ^a^	Fe	1.00 (1.00–1.00)	0.85	1.00 (1.00–1.00)	0.54	1.00 (1.00–1.01)	0.33
	Zn	1.01 (0.97–1.05)	0.57	1.03 (0.99–1.08)	0.13	1.04 (1.00–1.08)	0.08
	Cu	0.98 (0.96–1.01)	0.15	0.98 (0.96–1.01)	0.17	0.99 (0.97–1.00)	0.13
	Mg ^c^	0.91 (0.75–1.09)	0.33	1.01 (0.84–1.22)	0.89	1.07 (0.89–1.29)	0.47
	P	0.82 (0.55–1.23)	0.34	0.97 (0.65–1.46)	0.88	1.05 (0.73–1.53)	0.77
	Ca ^c^	0.93(0.77–1.11)	0.41	0.89 (0.73–1.07)	0.22	0.89 (0.74–1.07)	0.21
	Cu/Zn	0.81(0.42–1.57)	0.54	0.72 (0.37–1.39)	0.32	0.53 (0.27–1.05)	0.07
Gestational Diabetes ^a^	Fe	1.00 (1.00–1.00)	0.99	1.00 (1.00–1.01)	0.80	1.00 (1.00–1.01)	0.33
	Zn	1.00 (0.96–1.04)	0.83	1.03 (0.98–1.08)	0.25	1.04 (0.99–1.09)	0.12
	Cu	1.00 (0.97–1.02)	0.70	1.02 (0.99–1.04)	0.17	1.00 (0.99–1.02)	0.68
	Mg ^c^	0.98 (0.79–1.21)	0.84	1.12(0.90–1.40)	0.31	1.29 (1.03–1.62)	0.03
	P	1.12 (0.76–1.66)	0.55	1.59 (1.01–2.51)	0.04	1.76 (1.10–2.81)	0.02
	Ca ^c^	0.94 (0.77–1.14)	0.52	1.16 (0.93–1.46)	0.18	1.02 (0.84–1.26)	0.80
	Cu/Zn	1.17(0.63–2.16)	0.62	1.20 (0.77–1.88)	0.41	0.76 (0.44–1.31)	0.32
Obstetric Hemorrhage ^b^	Fe	1.00 (1.00–1.01)	0.35	1.00 (1.00–1.01)	0.57	1.00 (1.00–1.01)	0.39
	Zn	1.03 (0.98–1.08)	0.21	1.01 (0.96–1.07)	0.67	1.00 (0.94–1.06)	0.99
	Cu	1.00 (0.97–1.03)	0.95	1.01 (0.98–1.04)	0.67	0.99 (0.96–1.02)	0.56
	Mg ^c^	1.10 (0.83–1.44)	0.51	0.92 (0.70–1.19)	0.51	0.91 (0.70–1.18)	0.46
	P	0.81 (0.47–1.42)	0.48	0.61 (0.34–1.08)	0.09	0.72 (0.45–1.16)	0.18
	Ca ^c^	1.14 (0.88–1.47)	0.63	1.19 (0.91–1.58)	0.21	1.28 (0.94–1.74)	0.11
	Cu/Zn	0.48 (0.12–1.86)	0.29	0.77 (0.33–1.80)	0.54	1.21 (0.73–2.02)	0.46
Admission to Neonatal Intensive Care Unit ^a^	Fe	1.00 (1.00–1.01)	0.42	1.00 (1.00–1.00)	0.91	1.00 (0.99–1.00)	0.14
	Zn	1.06 (1.03–1.10)	<0.01	1.01 (0.98–1.06)	0.31	1.01 (0.98–1.05)	0.45
	Cu	1.00 (0.98–1.02)	0.91	1.00 (0.98–1.02)	0.89	1.00 (0.98–1.01)	0.47
	Mg ^c^	1.21 (1.00–1.46)	0.04	1.02 (0.86–1.20)	0.78	0.94 (0.79–1.11)	0.50
	P	1.68 (1.15–2.44)	0.01	1.07 (0.76–1.50)	0.71	0.96 (0.69–1.32)	0.80
	Ca ^c^	1.18(1.00–1.40)	0.05	1.29 (1.05–1.57)	0.02	1.28 (1.05–1.55)	0.02
	Cu/Zn	0.34 (0.13–0.84)	0.02	0.95 (0.62–1.46)	0.82	0.88 (0.58–1.33)	0.53
Low Birth Weight ^a^	Fe	1.00 (1.00–1.00)	0.59	1.00 (1.00–1.00)	0.74	1.00 (1.00–1.00)	0.98
	Zn	1.03 (1.00–1.06)	0.03	1.00 (0.97–1.04)	0.83	1.01 (0.98–1.04)	0.57
	Cu	1.00 (0.98–1.02)	0.83	1.01 (1.00–1.03)	0.09	1.01 (1.00–1.02)	0.19
	Mg ^c^	1.15 (0.98–1.36)	0.09	1.07 (0.92–1.25)	0.38	1.12 (0.96–1.31)	0.14
	P	1.01 (0.75–1.37)	0.95	0.91 (0.66–1.25)	0.58	1.05 (0.78–1.41)	0.73
	Ca ^c^	1.08 (0.93–1.25)	0.32	1.14 (0.97–1.20)	0.12	1.19 (1.01–1.41)	0.03
	Cu/Zn	0.63 (0.33–1.21)	0.16	1.19 (0.82–1.73)	0.36	1.11 (0.81–1.52)	0.39
Preterm Birth ^a^	Fe	1.00 (1.00–1.01)	0.18	1.00 (1.00–1.01)	0.17	1.00 (1.00–1.01)	0.24
	Zn	1.04 (1.01–1.07)	0.009	1.01 (0.98–1.04)	0.60	1.03 (1.00–1.06)	0.08
	Cu	1.00 (0.98–1.01)	0.54	1.00 (0.98–1.01)	0.74	1.00 (0.98–1.01)	0.50
	Mg ^c^	1.19 (1.01–1.40)	0.04	1.10 (0.95–1.28)	0.20	1.11 (0.96–1.28)	0.17
	P	1.20 (0.89–1.61)	0.24	1.08 (0.80–1.47)	0.61	1.29 (0.97–1.73)	0.08
	Ca ^c^	1.09 (0.94–1.26)	0.24	1.15 (0.98–1.34)	0.07	1.09 (0.94–1.26)	0.27
	Cu/Zn	0.45 (0.21–0.92)	0.03	0.52 (0.26–1.05)	0.07	0.76 (0.50–1.15)	0.19

OR = odds ratio; CI = confidence interval. ^a^ Models adjusted by age (years), glucose, triglycerides (mg/dL), cholesterol (mg/dL), 25-OH vitamin D (ng/dL), body mass index, multivitamin supplementation (yes or no), and socioeconomic status (low, medium, or high). ^b^ Models adjusted by age (years), glucose, triglycerides (mg/dL), cholesterol (mg/dL), 25-OH vitamin D (ng/dL), body mass index, and multivitamin supplementation (yes or no). ^c^ The metal ion levels were modeled in deciles.

**Table 4 ijms-25-13206-t004:** Longitudinal metal ion associations with adverse perinatal outcomes.

		Intercept		Slope	
Outcome	Metal Ion	OR (CI 95%)	*p* Value	OR (CI 95%)	*p* Value
Preeclampsia	Fe	1.28 (0.66–2.42)	0.45	1.66 (0.81–3.47)	0.17
	Zn	1.84 (0.94–3.61)	0.07	1.89 (0.92–4.00)	0.09
	Cu	0.72 (0.39–1.27)	0.28	0.74 (0.38–1.32)	0.33
	Mg	1.09 (0.49–2.41)	0.83	1.75 (0.76–4.03)	0.19
	P	0.85 (0.45–1.49)	0.58	1.21 (0.67–2.18)	0.52
	Ca	0.77 (0.46–1.28)	0.32	0.63 (0.29–1.22)	0.21
	Cu/Zn	1.30 (0.64–2.44)	0.43	0.41 (0.15–0.94)	0.07
Gestational Diabetes	Fe	1.40 (0.64–3.00)	0.39	1.53 (0.67–3.66)	0.32
	Zn	2.12 (0.96–4.70)	0.06	2.43 (1.08–5.95)	0.04
	Cu	0.89 (0.47–1.66)	0.73	1.25 (0.68–2.18)	0.45
	Mg	2.92 (1.11–8.40)	0.04	2.72 (1.14–6.85)	0.03
	P	1.79 (0.96–3.31)	0.06	1.97 (1.04–4.02)	0.04
	Ca	1.02 (0.58–1.79)	0.94	1.08 (0.57–1.89)	0.80
	Cu/Zn	1.31 (0.72–2.37)	0.36	0.65 (0.27–1.27)	0.27
Obstetric Hemorrhage ^a^	Fe	1.41 (0.59–3.24)	0.40	1.23 (0.53–2.99)	0.60
	Zn	1.16 (0.43–2.82)	0.80	0.76 (0.32–1.83)	0.50
	Cu	1.19 (0.55–2.39)	0.60	0.77 (0.30–1.67)	0.50
	Mg	0.67 (0.22–1.88)	0.50	0.70 (0.26–1.95)	0.50
	P	0.47 (0.17–1.12)	0.10	0.65 (0.31–1.33)	0.20
	Ca	1.46 (0.63–3.43)	0.40	1.57 (0.85–2.80)	0.10
	Cu/Zn	0.51 (0.15–1.29)	0.20	1.26 (0.63–2.19)	0.40
Admission to Neonatal Intensive Care Unit	Fe	0.93 (0.50–1.65)	0.82	0.60 (0.34–1.06)	0.08
	Zn	1.93 (1.04–3.64)	0.04	0.76 (0.41–1.40)	0.39
	Cu	1.12 (0.69–1.80)	0.64	0.73 (0.40–1.26)	0.28
	Mg	1.24 (0.62–2.48)	0.54	0.62 (0.31–1.20)	0.16
	P	1.73 (1.03–2.93)	0.04	0.65 (0.39–1.05)	0.08
	Ca	1.84 (1.11–3.16)	0.02	1.25 (0.81–1.92)	0.30
	Cu/Zn	0.58 (0.28–1.08)	0.11	0.97 (0.56–1.53)	0.91
Low Birth Weight	Fe	1.06 (0.62–1.77)	0.82	0.95 (0.56–1.63)	0.85
	Zn	1.33 (0.76–2.31)	0.31	0.85 (0.48–1.48)	0.57
	Cu	1.08 (0.68–1.69)	0.73	1.28 (0.83–1.94)	0.25
	Mg	1.65 (0.88–3.14)	0.12	1.36 (0.75–2.53)	0.32
	P	0.94 (0.58–1.45)	0.79	1.01 (0.66–1.56)	0.95
	Ca	1.23 (0.79–1.94)	0.35	1.40 (0.93–2.07)	0.10
	Cu/Zn	0.75 (0.43–1.20)	0.26	1.23 (0.80–1.83)	0.31
Premature Birth	Fe	1.06 (0.62–1.77)	0.82	0.95 (0.56–1.63)	0.85
	Zn	1.33 (0.76–2.31)	0.31	0.85 (0.48–1.48)	0.57
	Cu	1.08 (0.68–1.69)	0.73	1.28 (0.83–1.94)	0.25
	Mg	1.65 (0.88–3.14)	0.12	1.36 (0.75–2.53)	0.32
	P	0.94 (0.58–1.45)	0.79	1.01 (0.66–1.56)	0.95
	Ca	1.23 (0.79–1.94)	0.35	1.40 (0.93–2.07)	0.10
	Cu/Zn	0.75 (0.43–1.20)	0.26	1.23 (0.80–1.83)	0.31

Models adjusted by age (years), glucose, triglycerides (mg/dL), cholesterol (mg/dL), 25-OH vitamin D (ng/dL), body mass index, multivitamin supplementation (yes or no), and socioeconomic status (low, medium, or high). ^a^ Models adjusted by age (years), glucose, triglycerides (mg/dL), cholesterol (mg/dL), 25-OH vitamin D (ng/dL), body mass index, and multivitamin supplementation (yes or no). Abbreviations: OR = odds ratio; CI = confidence interval.

## Data Availability

The original contributions presented in this study are included in the article or Appendix A. Further inquiries can be directed to the corresponding authors.

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
