# Peer review of "Maternal Metal Ion Status Along Pregnancy and Perinatal Outcomes in a Group of Mexican Women"

_ijms, 2024, doi:10.3390/ijms252313206_

Round 1

Reviewer 1 Report

Comments and Suggestions for Authors

This prospective cohort study examined changes in metal ion levels during the three trimesters in 206 pregnant women from the OBESO cohort treated at the National Institute of Perinatal Medicine in Mexico City between 2017 and 2020. In addition, the study assessed the impact of pre-existing metabolic status on metal ion status and evaluated the association between maternal metal ion levels and perinatal outcomes. The study observed notable discrepancies in metal ion levels across trimesters, with iron and magnesium levels exhibiting a marked decline and copper levels demonstrating an increase as the pregnancy advanced. Maternal hypothyroidism was linked to significantly diminished zinc and magnesium levels during pregnancy. Regression analyses revealed a robust correlation between maternal metal ion levels and perinatal outcomes. Some suggested changes are provided below.

Major Revision:

1.       Perinatal outcomes of subjects may also be affected by factors such as level of education and family economic status. These important factors should also be taken into account.

2.       The results of association analyses can be validated against each other using a variety of hybrid exposure models such as WQS and q g-comp.

3.       Some results were overlooked and not described in the text, such as the protective effect of Fe in early pregnancy against prematurity and the harmful effects of P.

4.       Avoid using sentences such as ‘xxx increase/decrease risk’ throughout the text, such as line 238.

5.       It is recommended that additional sensitivity analysis be conducted to test the robustness of the primary analysis.

6.       It is suggested that the first paragraph of the discussion section provide a summary of the entire text.

Minor Revision:

1.       Figures and tables need to be double-checked for revisions.

a)    Please add notes below Table 1 about the abbreviations in the table.

b)    It should be noted that while some p-values in Table 3 are less than 0.05, they are not bolded. Conversely, some p-values greater than 0.05 are bolded. Therefore, it is necessary to exercise caution and verify the accuracy of the displayed data.

2.    It is recommended that serial numbers be added to the charts in the supplementary material and that they be aligned with the text.

3.    Please add a p-value to each subgraph of Figure S3.

4.    Line 118: “metabolic condition” change to hypothyroidism”.

5.    Please standardize your writing and pay attention to the format.

a) Line 244-246: “In this prospective cohort study, the observed variations in the levels of Cu, Fe, Zn, Mg, and P throughout pregnancy, supporting previous findings from studies conducted in Korean and Chinese”. The sentence lacks a predicate and does not cite the source of the quote.

b) Line 399: “mm Hg” change to “mmHg”.

c) The word “and” does not need to be preceded by a comma when joining two parallel phrases. This problem occurs several times in the text.

Reviewer 2 Report

Comments and Suggestions for Authors

This article helps to understanding the role of metal ions in pregnancy outcomes. The authors conducted a prospective cohort study on 206 pregnant women from Mexico City, examining changes in metal ion levels across the three trimesters and assessing their association with perinatal outcomes. 

Here are some specific suggestions,

1. In the introduction, it would be beneficial to provide a more detailed discussion on the roles of metal ions in maternal health and fetal development. For instance, include more information on how iron, zinc, copper, calcium, magnesium, and phosphorus deficiencies affect both mothers and their babies. This will help readers understand the significance of the research better.

2. Method section

Sample Selection Criteria Clarification: It would be helpful to describe the criteria for participant selection more explicitly, including how participants were determined to be healthy and what exclusion criteria were used. This will ensure the representativeness and reliability of the sample.

3. Results section

(1)Chart Explanations: The explanations for the figures need to be more detailed and specific. For example, in Figure 1, a detailed explanation of the trends in metal ion concentrations across different trimesters and their statistical significance should be provided. Also, explanations for significant differences should be more rigorous and detailed.

(2)Association Analysis Deepening: When discussing the association between metal ion levels and perinatal outcomes, additional details and examples should be included to support the conclusions. For instance, describe which metal ions are significantly associated with preterm birth, low birth weight, etc., and explain their possible biological mechanisms.

4. Discussion section

(1)Study Limitations Discussed More Thoroughly: In the discussion section, it is essential to discuss the limitations of the study and potential confounding factors more thoroughly. This could include limitations due to the observational nature of the study and how further interventional studies might validate these findings.

(2)Propose More Specific Future Research Directions: It is recommended to propose more concrete suggestions for future research directions in this section. 

The above suggestions are for the author's reference and improvement.

Reviewer 3 Report

Comments and Suggestions for Authors

You have included a large number of female subjects and obtained a large number of results.

You have presented well the changes during pregnancy for several analytes. 

However, diet and supplementation must also be taken into account for the interpretation of the results. These can significantly alter the result of a particular analyte. Do you have data on intake, diet?

To the greatest extent possible, you are presenting an association of the concentrations of the metals of interest with preterm birth or pre-eclampsia.

When it comes to predicting the risk of pre-eclampsia, it would be interesting to calculate the Cu/Zn ratio. Given that both parameters are very important, it is important to interpret each one separately as well as the ratio.

Among the results, you also presented the determination of vitamin 25(OH)D. Given that vitamin D plays an important role, I suggest that you calculate the association of the metal determination with vitamin D. In areas where vitamin D has seasonal variations due to latitude, the results should also be interpreted in the light of these differences.

Technically, Table 3 is poor and I suggest that you prepare it in a different format to make the results more visible.

Round 2

Reviewer 3 Report

Comments and Suggestions for Authors

I am satisfied with the answers and the additions to the text, as well as with the additional calculations.